# Outcomes of Radical Hysterectomy for Early-Stage Cervical Carcinoma, with or without Prior Cervical Excision Procedure

**DOI:** 10.3390/cancers16112051

**Published:** 2024-05-29

**Authors:** Dimitrios Nasioudis, Nayla Labban, Stefan Gysler, Emily M. Ko, Robert L. Giuntoli, Sarah H. Kim, Nawar A. Latif

**Affiliations:** Division of Gynecologic Oncology, Penn Medicine, University of Pennsylvania, Philadelphia, PA 19104, USA

**Keywords:** cervical cancer, hysterectomy, laparoscopes

## Abstract

**Simple Summary:**

The impact of a prior cervical excisional procedure on the oncologic outcomes of patients with early-stage cervical cancer undergoing radical hysterectomy is not established. Smaller retrospective studies suggest that conization prior to the performance of radical hysterectomy may be associated with lower risk of tumor relapse, especially for patients undergoing minimally invasive hysterectomy. We utilized a large hospital-based database and identified patients with FIGO 2009 stage IB1 disease who had primary surgical treatment. Approximately one in three patients had a prior excisional procedure performed within 3 months of radical surgery. We demonstrated that the performance of an excisional procedure prior to radical hysterectomy may be associated with better overall survival. For patients who had a prior excisional procedure, minimally invasive surgery was not associated with worse overall survival compared to laparotomy even after controlling for confounders.

**Abstract:**

Objective: To investigate the impact of a prior cervical excisional procedure on the oncologic outcomes of patients with apparent early-stage cervical carcinoma undergoing radical hysterectomy. Methods: The National Cancer Database (2004–2015) was accessed, and patients with FIGO 2009 stage IB1 cervical cancer who had a radical hysterectomy with at least 10 lymph nodes (LNs) removed and a known surgical approach were identified. Patients who did and did not undergo a prior cervical excisional procedure (within 3 months of hysterectomy) were selected for further analysis. Overall survival (OS) was evaluated following the generation of Kaplan–Meier curves and compared with the log-rank test. A Cox model was constructed to control a priori-selected confounders. Results: A total of 3159 patients were identified; 37.1% (n = 1171) had a prior excisional procedure. These patients had lower rates of lymphovascular invasion (29.2% vs. 34.9%, *p* = 0.014), positive LNs (6.7% vs. 12.7%, *p* < 0.001), and a tumor size >2 cm (25.7% vs. 56%, *p* < 0.001). Following stratification by tumor size, the performance of an excisional procedure prior to radical hysterectomy was associated with better OS even after controlling for confounders (aHR: 0.45, 95% CI: 0.30, 0.66). The rate of minimally invasive surgery was higher among patients who had a prior excisional procedure (61.5% vs. 53.2%, *p* < 0.001). For these patients, performance of minimally invasive radical hysterectomy was not associated with worse OS (aHR: 1.37, 95% CI: 0.66, 2.82). Conclusions: For patients undergoing radical hysterectomy, preoperative cervical excision may be associated with a survival benefit. For patients who had a prior excisional procedure, minimally invasive radical hysterectomy was not associated with worse overall survival.

## 1. Introduction

Cervical cancer is currently the third most common gynecologic malignancy in the United States [1]. According to major guidelines such as the National Comprehensive Cancer Network guidelines, for patients with FIGO 2019 IB2 disease as well as certain patients with IB1 disease, surgical management encompasses radical hysterectomy with lymphadenectomy or sentinel lymph node biopsy [2]. Minimally invasive techniques were initially extrapolated from the management of patients with endometrial cancer and aimed to minimize blood loss, decrease perioperative morbidity, and shorten inpatient hospital stay of patients undergoing radical hysterectomy [3]. The adoption of minimally invasive surgery for patients with cervical cancer precluded the generation of robust evidence on its oncologic safety. Nevertheless, in this patient group, a recent phase III randomized trial (LACC trial) demonstrated worse progression-free survival compared to laparotomy; disease-free survival rates at 4.5 years were 86% and 96.5%, respectively (HR: 3.74%, 95% CI: 1.63, 8.58) [4]. The negative impact of minimally invasive techniques on the oncologic outcomes of patients with cervical cancer was also demonstrated in several retrospective studies and database analyses [5,6]. Therefore, major guidelines recommend against the utilization of minimally invasive surgery in patients with early-stage cervical cancer.

To date, it is unclear whether worse oncologic outcomes are attributed to the utilization of uterine manipulator that can introduce tumor cells into the abdominal cavity, the intraperitoneal spread of tumor cells during the performance of colpotomy in the steep Trendelenburg position, or inadequate surgical technique [7]. There are emerging data from several retrospective studies that suggest that the performance of preoperative cold knife cone may have a protective effect through a reduction in tumor size, especially if negative margins are obtained [8,9,10,11,12,13,14,15,16,17,18,19,20]. However, given the small number of patients included in these studies, the heterogeneity of the population, and the inherent biases, conclusions cannot be drawn. The aim of the present study was to investigate the outcomes of patients with stage IB cervical carcinoma who underwent radical hysterectomy with or without a preoperative cervical procedure, using real-world data derived from a large hospital-based database.

## 2. Methods

We accessed the National Cancer Database (NCDB) and identified patients diagnosed between 2010 and 2015 who did not have a history of another tumor and were diagnosed with microscopically confirmed invasive cervical carcinoma. Based on ICD-O-3 histology codes, those with squamous, adenosquamous carcinoma, or adenocarcinoma histology were selected. Patients with FIGO 2009 stage IB1 disease who underwent minimally invasive (traditional laparoscopy or robotic-assisted laparoscopy) or open radical hysterectomy (based on site-specific surgery codes) with at least 10 lymph nodes removed (based on data derived from the pathology report) and had at least one month of follow-up were selected for further analysis. We excluded patients with unknown mode of surgery, rare histologic subtypes, and unknown disease stage, as well as those who received neoadjuvant radiation therapy or neoadjuvant chemotherapy. We identified patients who underwent another cervical surgical procedure up to 3 months before radical hysterectomy. In the National Cancer Database, the performance and interval of a surgical procedure (other than tumor biopsy) at the primary tumor site before a definitive surgical procedure are documented; however, exact surgical details are not available. Cervical biopsy is not considered a surgical procedure and is not recorded in the database.

Demographic, clinico-pathological, and treatment characteristics were extracted from the de-identified dataset. Based on average age of menopause, patient age was dichotomized into ≤50 and >50 years, while presence of medical comorbidities was assessed using the Charlson–Deyo comorbidity index and dichotomized into present (score ≥ 1) and absent (score 0). Insurance status was grouped into private, government (Medicare and Medicaid), and uninsured/unknown. Data on the margin and lymph node status as well as the presence of lymph–vascular invasion were derived from the pathology report. Distribution of categorical and continuous variables was compared with the chi-square and Mann–Whitney U tests, respectively. Following the generation of Kaplan–Meier curves, overall survival (OS) was assessed and further compared with the log-rank test. Overall survival was examined following stratification by tumor size (≤2 cm, >2 cm). A Cox model was constructed to control for a priori-selected confounders. All statistical analyses were performed with the Statistical Package for the Social Sciences v.29 (International Business Machines Corporation Armonk, New York, NY, USA), and the alpha level of statistical significance was set at 0.05.

The NCDB is a hospital-based database and captures approximately 70% of all malignancies diagnosed annually in the United States [21]. The American College of Surgeons and the Commission on Cancer have not verified and are not responsible for the analytical or statistical methodology employed, or the conclusions drawn from these data. The present study was deemed exempt from Institutional Board Review from Penn Medicine IRB (Protocol #829268).

## 3. Results

A total of 3159 patients who met the inclusion criteria were identified; 37.1% (n = 1171) underwent an excisional procedure prior to radical hysterectomy. In the present cohort, the rate of minimally invasive surgery (MIS) was 55.9% (n = 1766). Among patients who had MIS, 79.6% (n = 1407) underwent robotic-assisted and 20.4% (n = 359) traditional laparoscopy. The median patient age was 48 years (range: 21–86 years), and most patients were White (81.7%), without medical comorbidities (87.7%), and had private insurance (61%). The most common histologic subtype was squamous cell carcinoma (58.6%) and most tumors were ≤2 cm in size (n = 1450, 45.9%). Patients had a median of 19 lymph nodes removed (range: 10–80). Based on pathology report, 33% of tumors had LVSI, 10.5% had positive lymph nodes, and 3.5% had a positive resection margin. The rate of adjuvant radiation therapy in the present cohort was 23.2%.

Table 1 summarizes the clinico-pathological characteristics of the patient population stratified by receipt of prior excisional procedure. Patients who had an excisional procedure prior to radical hysterectomy were less likely to have lymphovascular invasion (LVSI) (29.2% vs. 34.9%, *p* = 0.014), positive lymph nodes (6.7% vs. 12.7%, *p* < 0.001), or tumors larger than 2 cm (25.7% vs. 56%, *p* < 0.001), and they were less likely to receive adjuvant radiation therapy (15.5% vs. 27.8%, *p* < 0.001). In addition, the rate of minimally invasive radical hysterectomy was higher among patients who underwent a prior excisional procedure (60.5% vs. 53.2%, *p* < 0.001). These patients were also younger (median: 43 vs. 44 years, *p* = 0.006) and more likely to have private insurance (65.6% vs. 58.4%, *p* = 0.001). The median number of LNs removed was comparable between the two groups (19 LNs in both, *p* = 0.87). However, the rate of lymph node metastases was higher among patients who did not have a prior excisional procedure (12.7% vs. 6.7%, *p* < 0.001). The rate of positive margins (3.2% vs. 3.7%, *p* = 0.50) was similar between the two groups.

According to the reverse Kaplan–Meier method, the median follow-up of the present cohort was 42.32 months. For patients with tumors ≤2 cm, those who had a prior excision (n = 703) had better OS compared to those who did not (n = 747) (*p* = 0.008 from log-rank test), and 5-year OS rates were 97.1% and 94.1%, respectively (*p* = 0.008) (Figure 1). Similarly, for patients with tumor size >2 cm, those who had a prior excisional procedure had better OS compared to those who did not (*p* = 0.004 from log-rank test), and 5-year OS rates were 89.9% and 82.5%, respectively (Figure 2). After controlling for confounders such as mode of radical hysterectomy (open vs. minimally invasive), tumor size (≤2 cm vs. >2 cm), histology, presence of LN metastases, lymphovascular invasion, patient age, insurance status, and the presence of medical comorbidities, patients who underwent a prior excisional procedure had better OS (aHR: 0.45, 95% CI: 0.30, 0.66).

For patients who had a prior excisional procedure, there was no difference in OS between the MIS (n = 709) and open (n = 462) radical hysterectomy (*p* = 0.45), and 5-year OS rates were 94.6% and 95.7%, respectively (Figure 3), even after controlling for confounders (aHR: 1.37, 95% CI: 0.66, 2.82). However, for patients who did not undergo an excisional procedure prior to radical hysterectomy, performance of minimally invasive surgery (n = 1057) was associated with worse OS compared to laparotomy (n = 931) (*p* = 0.001), and 5-year OS rates were 84.9% and 90.2% (Figure 4), even after controlling for the aforementioned confounders (aHR: 1.91, 95% CI: 1.39, 2.62).

## 4. Discussion

We utilized real-world data derived from a large national hospital-based database and demonstrated that for patients with FIGO 2009 stage IB1 cervical carcinoma, the performance of an excisional procedure prior to radical hysterectomy may be associated with better overall survival even after controlling for important confounders. Interestingly, when examining patients who underwent preoperative excisional procedure, there was no difference in overall survival between the minimally invasive and open radical hysterectomy groups.

There are emerging data from retrospective studies that suggest that preoperative conization may have a positive impact on the oncologic outcomes of patients with early-stage cervical cancer undergoing radical hysterectomy and may be associated with a lower risk of tumor relapse. A multicenter retrospective study that employed propensity score-matching statistical analysis included 332 patients with FIGO 2009 stage IB1 disease undergoing radical hysterectomy. Patients who had preoperative conization were more likely to undergo minimally invasive surgery (88.5% vs. 68.2%, *p* < 0.001) and less likely to receive adjuvant treatment (27.7% vs. 41%, *p* = 0.015). Conization was associated with better disease-free survival (89.8% vs. 80%, *p* = 0.01) and a trend towards better overall survival (97.1% vs. 91.4%, *p* = 0.11) [8]. Similarly, a recent multicenter study from South Korea included data from 593 patients with FIGO 2009 stage IB1-IIA2 disease who underwent radical hysterectomy and had a median follow-up of 114.8 months. Again, for patients with IB1 disease who underwent preoperative MRI, performance of conization (performed in 39.3% of patients) was identified by multivariable analysis as a favorable prognostic factor (HR: 0.32, 95% CI: 0.15, 0.69) [11]. A benefit was found in patients with stage IB1 disease regardless of tumor size (HR: 0.34, 95% CI: 0.14, 0.86) but, following stratification by tumor size, not for patients with tumors ≤2 cm in size [11]. In an analysis of the SUCCOR multicenter study, which included a total of 1156 patients with FIGO 2009 stage IB1 cervical cancer undergoing radical hysterectomy between 2013 and 2014 in 126 institutions, 733 (63.4%) had undergone a preoperative cold knife cone [12]. The majority (77.5%, n = 145) had positive margins on the cone specimen, while 80.7% (n = 151) had residual tumor on the final hysterectomy specimen. The authors performed propensity score matching to control for baseline demographic and pathologic characteristics and reported a 65% and 75% reduction in the risk of relapse and death, respectively, among patients who had a preoperative cold knife cone after a median follow-up of 58 months [12]. However, information on the indication of conization and the surgeon’s specialty was not collected. Another multicenter retrospective study from South Korea included 1254 patients with FIGO 2009 stage IB disease who underwent radical hysterectomy between 2006 and 2021 and had negative margins and lymph nodes. Following propensity score matching, the authors found a protective effect of conization in recurrence rates limited to patients with tumors larger than 2 cm who underwent minimally invasive surgery [14]. In a single-institution study from China, among 1273 patients with FIGO 2018 stage IB1 disease (tumor size ≤ 2 cm), the rate of prior conization was 51.6% [15]. Residual disease in the hysterectomy specimen was present in 64.6% of patients. After a median follow-up of 50.3 months, 2.4% of patients recurred [15]. By multivariable analysis, the performance of conization was associated with lower probability of tumor relapse (HR: 0.26, 95% CI: 0.10, 0.63) as well as improved 5-year recurrence-free (98.6% vs. 95.1%, *p* = 0.001) and overall (99.2% vs. 97.2%, *p* = 0.035) survival [15]. A recent systematic review of the literature and meta-analysis identified 11 studies reporting on 4184 patients who underwent radical hysterectomy; 2122 patients underwent a prior cervical excisional procedure (n = 2122) [19]. Based on 1616 patients, prior conization was associated with better disease-free survival (HR: 0.23, 95% CI: 0.12, 0.44), while based on 1835 patients it was also associated with better overall survival (HR: 0.54, 95% CI: 0.33, 0.86) [19]. Based on data from these studies, an oncologic benefit was more evident in patients with smaller tumors and those who underwent minimally invasive surgery; however, it should be underlined that a formal synthesis of the subgroup results was not performed, given the small number of patients [19].

Tumor seeding, either from uterine manipulator placement or tumor exposure to the intraperitoneal cavity during the performance of colpotomy, is hypothesized to be one of the causes of increased risk of relapse with minimally invasive radical hysterectomy. A recent systematic review and meta-analysis that examined patterns of recurrence included data on 7626 patients derived from 22 studies and found that, compared to the open approach, minimally invasive radical hysterectomy was associated with a higher risk of peritoneal carcinomatosis (OR: 1.90, 95% CI: 1.32, 2.74) [22]. Protective maneuvers such as avoidance of uterine manipulator, closure of the vaginal cuff before colpotomy, and preoperative conization can eliminate or mitigate the risk of peritoneal tumor seeding. Based on two systematic reviews and meta-analyses of observational studies, the omission of a uterine manipulator alone may not fully mitigate this risk; however, protective colpotomy techniques are associated with superior oncologic outcomes [23,24].

Similar to our results, for patients with minimal or no residual disease on their preoperative imaging following a cervical excisional procedure or final hysterectomy specimen procedure, data from retrospective studies suggest that minimally invasive surgery may not be associated with worse survival outcomes. In the SUCCOR multicenter study, among patients with prior conization, those who had minimally invasive surgery did not have a worse relapse rate compared to those who underwent laparotomy (HR: 1.94; 95% CI: 0.49, 7.76; *p* = 0.35) [12]. Similarly, another multicenter retrospective study identified a total of 243 patients managed in institutions within the US who underwent conization and did not have any residual tumor in their preoperative imaging. For these patients, the relapse rate was comparable between the laparotomy (n = 72) and minimally invasive radical hysterectomy groups (n = 171) (1.4% vs. 2.9%, *p* = 0.48) [13]. In another multicenter study from Canada that included 238 patients with no residual carcinoma on the hysterectomy specimen, recurrences were exceedingly rare, occurring in only two patients [25]. Of note, in that study, 46.2% of patients had stage IB disease, while 81.9% underwent radical hysterectomy [25]. In a retrospective study examining patients who underwent minimally invasive surgery at three gynecologic oncology referral centers in Italy, Casarin et al. included a total of 186 patients with FIGO 2009 stage IA1-IB1 disease who had radical hysterectomy [9]. The performance of preoperative conization was associated with an overall lower risk of relapse (1.1% vs. 16.1%, *p* < 0.001) [9]. Although patients who had preoperative conization had smaller tumors, when examining patients with FIGO 2009 stage IB1 disease, preoperative conization was again associated with a lower risk of tumor relapse (1.8% vs. 17.2%, *p* = 0.004) [9]. Similarly, in another study following propensity score matching among patients who had minimally invasive radical hysterectomy (n = 397), the performance of preoperative conization (n = 256) was associated with better progression-free (5-year rates: 91.2% vs. 82.7%, *p* = 0.021) and overall (5-year rates: 96.9% vs. 93.2%, *p* = 0.066) survival [26]. In another retrospective study with a long follow-up, which included patients with stage IB disease and a tumor size ≤2 cm, those who underwent minimally invasive surgery without prior conization (n = 62) had increased risk of relapse compared to those who underwent prior conization (n = 94) [27]. Following multivariable analysis and adjusting for confounders, for patients who had prior conization and a tumor size ≤2 cm, there was no difference in time to recurrence between those who had laparotomy and minimally invasive surgery (HR: 1.85, 95% CI: 0.47, 7.35) [27]. In an exploratory analysis of the randomized LACC trial focusing on patients who had a preoperative conization, no difference in disease-free survival was identified between the laparotomy (n = 136) and minimally invasive groups (n = 128) (HR: 1.27; 95% CI: 0.39, 4.17; *p* = 0.69). However, similar to our study, when evaluating patients who did not undergo a preoperative conization, those who underwent minimally invasive surgery (n = 191) had a higher risk of tumor recurrence (19.4% vs. 4.5%) and worse disease-free survival (HR: 5.85; 95% CI: 2.47, 13.9; *p* < 0.001) compared to those who had laparotomy (n = 176) [28]. In the recent phase III randomized trial (SHAPE) that compared the oncologic safety of simple hysterectomy to standard-of-care radical hysterectomy and included 700 patients with low-risk early-stage cervical cancer (size ≤2 cm and limited stromal invasion), the majority (77%) of patients underwent minimally invasive surgery [29]. Although not a primary outcome of the trial, exploratory analyses revealed that there was no difference in risk of recurrence between the open (3.8%) and minimally invasive (3.6%) surgery groups [30]. However, approximately 561 (80.1%) patients underwent a preoperative conization or LEEP. Interestingly, no extrapelvic recurrence or death was observed in patients who had conization/LEEP with negative margins (n = 174) [30].

A major strength of our study was the large number of patients derived from a multicenter hospital-based database reflecting real-world data and permitting meaningful survival and subgroup analyses. However, several limitations of the present study should be mentioned. Firstly, given the lack of central pathology reports, possible tumor histology and stage misclassifications cannot be excluded. In addition, cone or LEEP specimen margins and residual tumor volume on preoperative imaging are not collected in the National Cancer Database. Moreover, while all patients had a surgical procedure other than cervical biopsy, preoperative indication and specific surgical details (e.g., performance of cold knife cone or loop electrode excision) were not available, and the specialty of the surgeon performing the cervical excision procedure (general gynecologist or gynecologic oncologist) is not reported. Also, details on the minimally invasive radical hysterectomy technique and whether a uterine manipulator was utilized or colpotomy protective techniques were performed are not available. Data on perioperative complications and estimated blood loss were not collected. More importantly, data on the presence and size of residual tumor at the hysterectomy specimen were not available. Lastly, the National Cancer Database does not include information on tumor relapse or the location of recurrence, precluding us from analyzing the impact of preoperative conization on the progression-free survival and patterns of relapse.

## 5. Conclusions

In the largest cohort to date, for patients with FIGO 2009 stage IB1 cervical carcinoma undergoing radical hysterectomy, the performance of a prior cervical excision procedure was associated with a better overall survival. For patients with a prior excisional procedure, the performance of minimally invasive radical hysterectomy did not confer a worse overall survival. Like other retrospective studies, the results of the present investigation should be regarded as hypothesis-generating. Since the excision of gross cervical lesions is not recommended based on current guidelines, it is possible that the performance of a conization procedure is a surrogate marker of a less aggressive biologic tumor behavior or favorable tumor characteristics such as smaller tumor size [31]. Further research is required to investigate whether patients with minimal or no visible disease following an excisional procedure can be safely offered a minimally invasive approach without jeopardizing oncologic outcomes, especially if protective maneuvers are incorporated in the surgical technique. Data from ongoing randomized trials exploring the role of minimally invasive surgery for patients with early-stage cervical cancer may provide additional evidence of a potential protective role of preoperative conization [32,33].

## Figures and Tables

**Figure 1 cancers-16-02051-f001:**
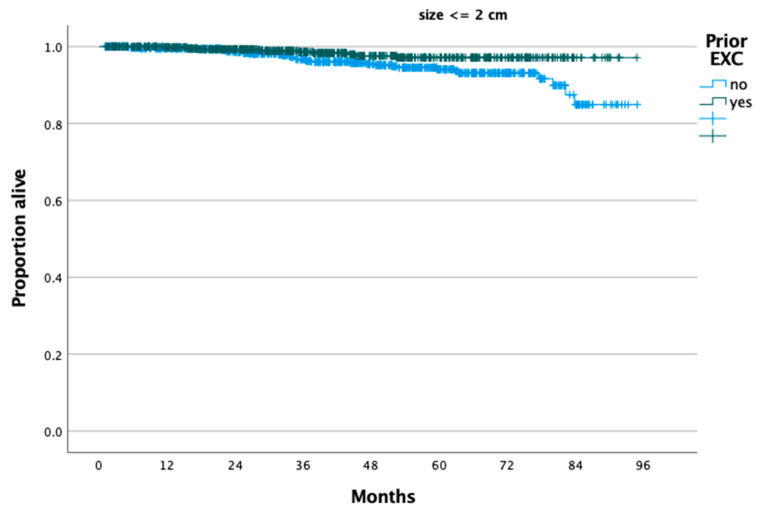
Overall survival of patients with stage IB cervical carcinoma and tumor size ≤2 cm who underwent radical hysterectomy stratified by performance of a prior cervical excision procedure.

**Figure 2 cancers-16-02051-f002:**
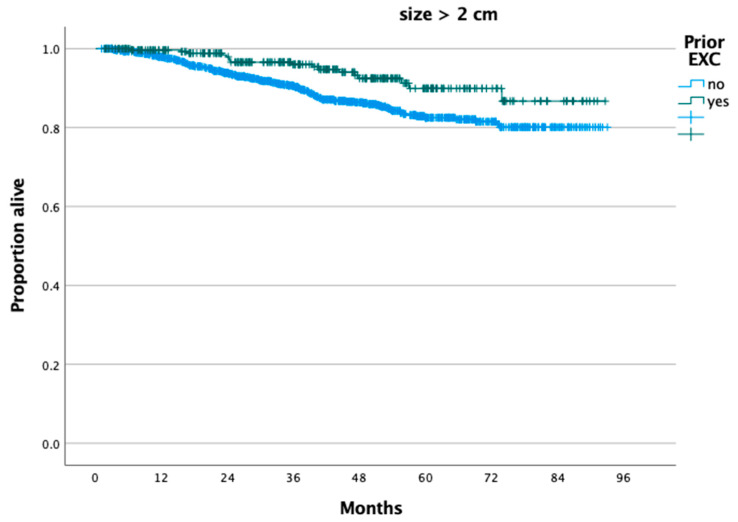
Overall survival of patients with stage IB cervical carcinoma and tumor size >2 cm who underwent radical hysterectomy stratified by performance of a prior cervical excision procedure.

**Figure 3 cancers-16-02051-f003:**
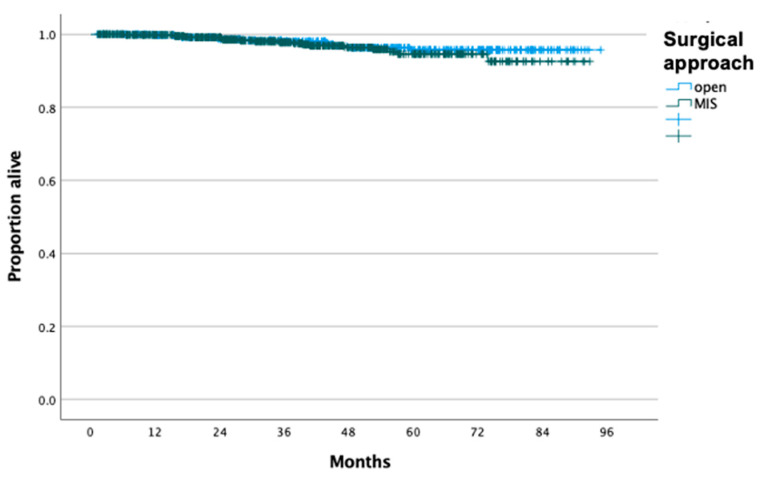
Overall survival of patients with stage IB cervical carcinoma who underwent radical hysterectomy and prior excisional procedure stratified by surgical approach.

**Figure 4 cancers-16-02051-f004:**
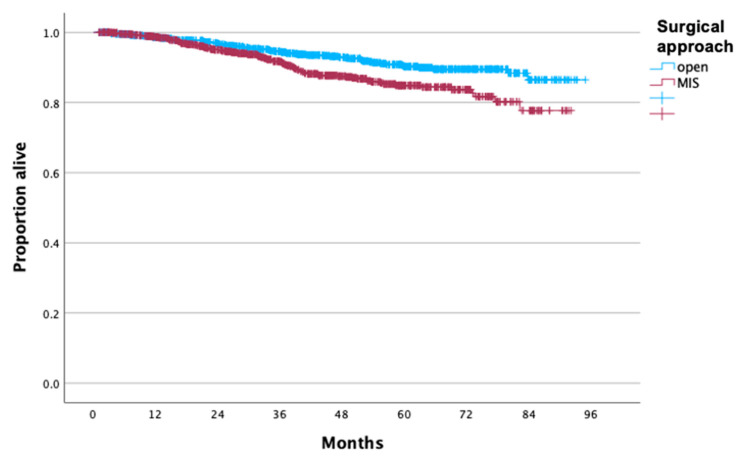
Overall survival of patients with stage IB cervical carcinoma who underwent radical hysterectomy without a prior excisional procedure stratified by surgical approach.

**Table 1 cancers-16-02051-t001:** Demographic and clinico-pathologic characteristics stratified by performance of prior excisional procedure.

	No Prior EXC	Prior EXC	*p*-Value
Age (years)			0.013
≤50	1356 (68.2%)	848 (72.4%)	
>50	632 (31.8%)	323 (27.6%)	
Race			0.20
White	1611 (81%)	969 (82.7%)	
Black	209 (10.5%)	100 (8.5%)	
Other/Unknown	168 (8.5%)	102 (8.7%)	
Comorbidities			0.064
No	1726 (86.8%)	1043 (89.1%)	
Yes	262 (13.2%)	128 (10.9%)	
Insurance			0.001
Private	1160 (58.4%)	768 (65.6%)	
Government	656 (33%)	327 (27.9%)	
Uninsured/Unknown	172 (8.6%)	76 (6.5%)	
Mode of surgery			<0.001
Open	931 (46.8%)	462 (39.5%)	
MIS	1057 (53.2%)	709 (60.5%)	
Tumor size			<0.001
≤2 cm	747 (37.6%)	703 (60%)	
>2 cm	1114 (56%)	401 (25.7%)	
Unknown	127 (6.4%)	167 (14.3%)	
Radiotherapy			<0.001
Yes	552 (27.8%)	181 (15.5%)	
No	1420 (71.4%)	980 (83.7%)	
Unknown	16 (0.8%)	10 (0.8%)	
Histology			0.047
Squamous	1164 (58.6%)	688 (58.8%)	
Adenosquamous	121 (6.1%)	48 (4.1%)	
Adenocarcinoma	703 (35.4%)	435 (37.1%)	
LVSI			0.014
No	1129 (56.8%)	723 (61.7%)	
Yes	693 (34.9%)	350 (29.9%)	
Unknown	166 (8.3%)	98 (8.4%)	

## Data Availability

Data are available following application to the American College of Surgeons.

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
