# Peer review of "Outcomes of Radical Hysterectomy for Early-Stage Cervical Carcinoma, with or without Prior Cervical Excision Procedure"

_cancers, 2024, doi:10.3390/cancers16112051_

Round 1

Reviewer 1 Report

Comments and Suggestions for Authors

This multicentric, retrospective, large cohort study evaluates the outcomes of radical hysterectomy for carcinoma of the cervix at FIGO stage IB with and without prior cervical excision procedure.

The paper is well written and the English language is appropriate and understandable.

The clinical arguments presented are quite interesting.

Data from this study try to provide some evidence of the potential protective role of pre-operative conization before radical surgery, especially in the case of a minimally invasive approach. 

However, it couldn’t provide strong scientific support to recommend the excision of gross cervical lesions routinely. Although the statistical analysis was well performed, this is a retrospective study with multiple biases that limit its clinical validity.

So far, the performance of a prior conization in not based on current guidelines even though it could be a potential surrogate marker of a less aggressive biologiccancer behaviour.

In any case, the limitations and bias of this study are correctly reported by the Authors.

Specific comments: 

This study utilized real-world data derived from a large hospital-based database. The Authors could present a Figure on “Study Population” showing women with data in the National Cancer Database who met inclusion criteria.

Of the women who underwent minimally invasive surgery, how many patients received robot-assisted laparoscopy instead of traditional laparoscopy?

Did the Authors explore whether the different minimally invasive approaches played a potential role in oncologic outcomes?

Was the rate of any intraoperative complications different in the two respective groups (prior conization vs any surgical cervical excision)?

Could the Authors give data on the recurrences (locally, ie vaginal vault or pelvis, vs distant relapses)? 

We suggest introducing “Surgical approach” instead of “Mode of surgery” in Figures 3 and 4.

HR and CI values could be included in the Figures.

Author Response

Comment 1: Of the women who underwent minimally invasive surgery, how many patients received robot-assisted laparoscopy instead of traditional laparoscopy?

Reply to comment 1: Among patients who had MIS, 79.6% (n=1407) underwent robotic-assisted and 20.4% (n=359) traditional laparoscopy.  (line 190-191)

Comment 2: Did the Authors explore whether the different minimally invasive approaches played a potential role in oncologic outcomes?

Reply to comment 2: there was no difference in OS between robotic and traditional laparoscopy (4-yr OS rates 91.1% vs 92.0%, p=0.56. 

Comment 3: Was the rate of any intraoperative complications different in the two respective groups (prior conization vs any surgical cervical excision)?

Reply to comment 3: the NCDB does not collect data on intraoperative or perioperative morbidity (lines 819-820)

Comment 4: Could the Authors give data on the recurrences (locally, ie vaginal vault or pelvis, vs distant relapses)? 

Reply to comment 4: the National Cancer Database does not include information on tumor relapse or location of recurrence, precluding us from analyzing the impact of preoperative conization on the progression-free survival and patterns of relapse (line 821-824)

Comment 5: We suggest introducing “Surgical approach” instead of “Mode of surgery” in Figures 3 and 4.

Reply to comment 5: changed

Comment 6: HR and CI values could be included in the Figures.

Reply to comment 6: HR and CI following adjustment for confounders are provided in the results section. 

Reviewer 2 Report

Comments and Suggestions for Authors

The results confirm previous findings regarding the protective role of conization in patients with early stages cervical cancer submitted to radical hysterectomy. A bias of study results might be a less advanced stage and a smaller tumor in previous conization arm.

Author Response

Comment 1: A bias of study results might be a less advanced stage and a smaller tumor in previous conization arm.

Reply to comment 1: although we control for tumor size and other confounders, as discussed in the conclusion we agree with the reviewer that prior conization may be surrogate of less aggresive biologic behavior or favorable pathologic characteristics (lines 830-831)